

# Enhancement of loop-mediated isothermal amplification (LAMP) with guanidine hydrochloride for the detection of *Streptococcus equi* subspecies *equi* (Strangles)

Alexandra Knox and Travis Beddoe

Department of Animal, Plant and Soil Science, La Trobe University, Bundoora, Victoria, Australia

## ABSTRACT

*Streptococcus equi* subspecies *equi*, commonly referred to as "strangles", poses a significant biosecurity challenge across equine farms worldwide. The continuous prevalence and highly transmissibility of strangles necessitates a rapid and accurate diagnostic procedure. However, current "gold-standard" techniques, such as cultures and quantitative polymerase chain reaction (qPCR), are unreliable or inaccessible, and require lengthy periods between sample collection and results. Moreover, the lack of a standardized detection protocol can lead to variations in results. This study aimed to develop a reproducible and field-deployable diagnostic assay to detect strangles in real-time. Utilising the rapid technique loop-mediated isothermal amplification (LAMP), we developed an assay targeting a conserved region of the *S. equi*-specific M gene (SeM). Additionally, we optimised our assay with guanidine hydrochloride (GuHCl) to enhance the assay's performance and detection capabilities. The Str-LAMP was able to detect *S. equi* within 13 minutes and 20 seconds for both synthetic DNA and clinical isolates, with a limit of detection (LOD) of 53 copies/µl. Our assay demonstrated high repeatability with the inter-coefficient of variation ranging from 0.17% to 3.93%. Furthermore, the clinical sensitivity and specificity was calculated at 91.3% and 93.3%, respectively, with a correct classification rate of 91.8%. The implementation of this newly developed strangles assay can be employed as an efficient aid for in-field surveillance programs. The assay's reproducibility can allow for equine managers to undertake routine self-surveillance on their properties, without the requirement of specialised training. The Str-LAMP assay has the potential to be a valuable tool to help mitigate potential strangles outbreaks.

Corresponding author
Travis Beddoe,
t.beddoe@latrobe.edu.au

## INTRODUCTION

Acute respiratory diseases in horses presents a significant biosecurity challenge in the equine industry worldwide. *Streptococcus equi* subspecies *equi*, the causative agent of strangles, is one of the most contagious respiratory diseases in equine (*Boyle et al., 2018*).

Strangles is currently endemic worldwide, excluding Ireland, causing large outbreaks that are challenging to overcome (*Bjornsdottir et al., 2016*; *Mitchell et al., 2021*). An infected horse can quickly become a chronic asymptomatic carrier which periodically spread the bacterium, therefore, become a source of new or recurrent outbreaks. Whilst some predisposing factors of disease susceptibility can be maintained, such as overcrowding and inadequate quarantine compliance, others are difficult to limit, such as weaning and severe weather (*Boyle et al., 2018*). In Australia, strangles is notifiable in only Victoria and New South Wales (*El-Hage et al., 2019*), and does not appear on the World Organisation for Animal Health (WOAH, formerly, Office International des Épizooties (OIE)) list of equine diseases despite literature debating this choice (*Mitchell et al., 2021*). Consequently, an estimation of strangles prevalence in Australia is limited (*El-Hage et al., 2019*). However, this does not diminish the impact on equine health and welfare, and the financial burden of an outbreak (*Boyle, 2023*). In fact, *Mitchell et al. (2021)* estimated some outbreaks involving hundreds of horses reportedly costs over AUD$ 580,000, a significant economic burden of large facilities.

Whilst *S. equi* disease can occur at any age, the most severe clinical manifestations are often seen in younger horses as the severity of the disease is dependent on the immune status (*Boyle et al., 2018*; *Pusterla et al., 2011*). Infected horses typically present with sudden pyrexia as the first sign, followed by a decrease in appetite or reluctancy to drink as a result of pharyngitis (*Boyle, 2023*). Abscesses are frequently formed around the submandibular and retropharyngeal lymph nodes. The progression of abscessation combined with pharyngitis may constrict the upper respiratory tract which can result in laboured breathing, hence the term ''strangles'' (*McLinden et al., 2023*; *Rendle et al., 2021*). Abscesses will eventually rupture and ooze thick purulent discharge through the nasal cavity. This discharge allows for shedding of the bacteria, contributing to the highly transmissible state of *S. equi* (*Boyle, 2023*; *McLinden et al., 2023*). Furthermore, an infected horse will continue to shed the infectious bacteria for at least 6 weeks post symptom cessation, during which time equine facilities must remain hypervigilant of *S. equi* transmission (*Rendle et al., 2021*).

Historically, microbial culture has been the ''gold-standard'' detection method for strangles. However, a low clinical sensitivity of just 30–40% and potential for false-negative results brought this technique into question (*Lindahl et al., 2013*). Subsequently diagnostic approaches for the detection of strangles are favoured towards nucleic acid amplification techniques, with multiple polymerase chain reaction (PCR)-based assays being developed. Currently, quantitative PCR (qPCR) is considered the test of choice for strangles diagnosis. However, there is no nationally standardised procedure regarding this diagnostic test, therefore, results may vary across laboratories (*Boyle et al., 2018*). While PCR marked a groundbreaking advancement, catalysing progress in molecular-based diagnostics, recent innovations in field-deployable technology have eclipsed its prominence. Among these advancements is loop-mediated isothermal amplification (LAMP), which is quickly overhauling conventional methods, particularly in equine diagnostics (*Boyle et al., 2018*; *Knox, Zerna & Beddoe, 2023*). LAMP utilises four to six primers that target a gene of interest at different regions alongside a strand displacing polyermase, enabling rapid and specific amplification. The ability to utilise a constant temperature, thus requiring only

a heat source, and capacity for visualisation of results in real-time allows for LAMP to be performed entirely in-field, greatly reducing the time between sample collection and results (*Notomi et al., 2000*). Furthermore, advancements in this technique have led to the incorporation of chemical additives to increase assay kinetics to outperform PCR-based methodologies (*Özay & McCalla, 2021*).

Our research aimed to develop an alternative LAMP assay to detect the *S. equi*-specific M protein (SeM) with the use of guanidine hydrochloride (GuHCl). A main purpose of this project was to demonstrate the ability of GuHCl in LAMP assays to decrease the time to positive and inter-coefficient of variation and decrease the limit of detection (LOD), whilst maintaining no adverse effects on the assay itself. This new Str-LAMP assay has the ability to be formatted into a simple results output to aid in routine surveillance of strangles across farms in Australia, delivering a rapid stable-side solution from current detection techniques.

## MATERIALS & METHODS

### Str-LAMP primer design

Three sets of primers were designed to target different regions of the *S. equi*-specific M gene (SeM) (Table 1), to allow for discrimination against closely related pathogen *S. zooepidemicus* (*Mitchell et al., 2021*; *Rotinsulu et al., 2023*). Primers were designed using PrimerExplorer (V5; Eiken Chemical Co. Ltd, Tokyo, Japan) with default settings, and were ordered through Bioneer Pacific (Daejeon, South Korea).

### Str-LAMP synthetic positive control design and preparation

A synthetic positive control was designed for assay optimisation and validation, to cover 353 base pairs of the SeM gene (Fig. 1), encompassing each primer set target region. A pBHA plasmid (Bioneer Pacific) was ordered containing the synthetic DNA insert and was transformed into DH5$\alpha$ *Escherichia coli* cells. Following plasmid purification using FastGene® Plasmid Mini kit (Nippon Genetics Europe EmBH, Düren, Germany), the synthetic positive control was amplified *via* PCR using the F3 and B3 from each primer set as the forward and reverse primers, respectively. Each reaction contained 50 µl of Go-Taq® Green mastermix (Promega, Madison, WI, United States), 0.8 µM of F3 and B3 of respective Str-LAMP primer sets (Table 1), and 20 ng/µl of the synthetic positive control, adjusted to 100 µl with dH$_2$O. Thermocycling was performed under the following conditions: 95 °C for 5 min, followed by 40 cycles of 95 °C for 1 min, 50 °C for 1 min, and 72 °C for 40 s, and a single cycle of 72 °C for 2 min. Amplicons were resolved on a 2% ethidium bromide (0.5 µg/ml) agarose gel for 40 min at 100 amps and were purified using NucleoSpin® PCR Clean Up kit (Takara Bio, Inc, Shiga, Japan). The purified linear PCR product was then used as a template for optimisation of the Str-LAMP assay.

### Str-LAMP primer optimisation

The performance of each primer set was assessed using the synthetic positive control as a template. Each reaction contained 15 µl of ISO-DR004 mastermix (OptiGene Limited, Horsham, United Kingdom), 0.2 µM of F3 and B3, 0.8 µM of FIP and BIP, and 0.4 µM of

**Table 1** **Primer sets designed for the Str-LAMP assay targeting the SeM gene.** Primer positions are based on GenBank accession number: U73162.

| Label | Sequence (5′–3′) | 5′pos | 3′pos | Length (bp) |
|---|---|---|---|---|
| Set1_F3[a] | GAAAACTAAGTGCCGGTGCA | 376 | 395 | 20 |
| Set1_B3[b] | AGCCCTTGCTGAATCAAGAC | 604 | 623 | 20 |
| Set1_FIP[c] | TCTTGGAGTCGCCGTACGACTAAAGTGTGTTGG GAGGGACAA | 416 (F2) | 435 (F2) | 42 |
| | | 458 (F1c) | 479 (F1c) | |
| Set1_BIP[d] | AAGTGGAGATGCCTCATCAGCCGCCTGTAAATC CCCAACAGA | 573 (B2) | 592 (B2) | 42 |
| | | 534 (B1c) | 542 (B1c) | |
| Set2_F3 | AACTCTGAGGTTAGTCGTAC | 477 | 466 | 20 |
| Set2_B3 | ATCATCTCTACCATACGCAG | 622 | 641 | 20 |
| Set2_FIP | GGCATCTCCACTTATGGCTATATCCGACTCCAAG ATTATCGCG | 496 (F2) | 487 (F2) | 43 |
| | | 510 (F1c) | 533 (F1c) | |
| Set2_BIP | TCATCAGCCCAAAAAGTTCGAAAGCTGAATCAAG ACCTCTCAA | 597 (B2) | 616 (B2) | 43 |
| | | 543 (B1c) | 556 (B1c) | |
| Set2_LB[e] | GCCTCTGTTGGGGATTTACAG | 570 | 590 | 20 |
| Set3_F3 | GCGATATAGCCATAAGTGGAG | 508 | 528 | 21 |
| Set3_B3 | ATTGTCTTCTATCCCCATCAG | 688 | 708 | 21 |
| Set3_FIP | TGCCTGTAAATCCCCAACAGAGGGCCTCATCAGCC CAAAAAGTTC | 532 (F2) | 552 (F2) | 45 |
| | | 571 (F1c) | 593 (F1c) | |
| Set3_BIP | AGAGGTCTTGATTCAGCAAGGGCACATCGATGAA AGGTGCATCA | 655 (B2) | 675 (B2) | 44 |
| | | 622 (B1c) | 622 (B1c) | |

**Notes.**
[a] F3, forward outer primer
[b] B3, backwards outer primer
[c] FIP, forward inner primer
[d] BIP, backwards inner primer
[e] LB, loop backwards

LB, with $5.3 \times 10^9$ copies/µl and $5.3 \times 10^8$ copies/µl (1 ng/µl and 0.1 ng/µl, respectively) of the synthetic positive control, or 5 µl of dH$_2$O for a no template control (NTC). Reactions were performed in triplicates using the Genie® II machine (OptiGene Limited) for 60 min of amplification at 65 °C followed by an anneal temperature of 98 °C to 80 °C, with a ramping rate of 0.05 °C per second. Reactions were considered positive if a time to positive (Tp) was achieved within 20 min, with a fluorescence ratio threshold at 0.02 normalised fluorescence units Reactions which amplified after the 20-minute threshold were considered negative, as per the manufacturer's guidelines (OptiGene Limited). The optimal primer set was selected based on the fastest average Tp.

After acquiring the optimal primer set, a series of assays were conducted with different concentrations of FIP, BIP, and LB, to determine the most effective primer concentration. Reactions were performed and assessed on the Genie® II machine as above, with 30 min

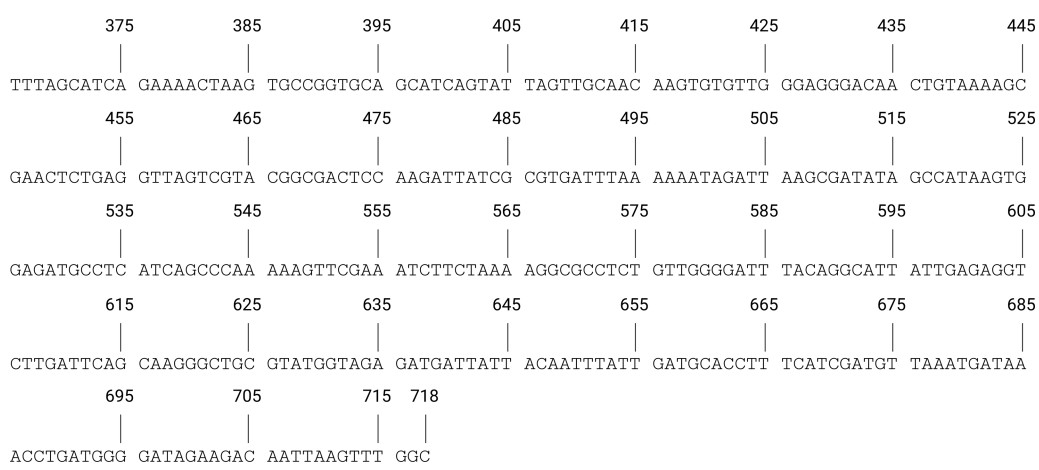

**Figure 1 The designed *Streptococcus equi* subspecies equi synthetic positive control.** Genome positioning according to GenBank accession number: U73162.

of amplification, using primer concentration of 0.2 μM for F3 and B3, and ranges of 0.8 to 1.6 μM of FIP and BIP, and 0.4 to 2.0 μM of LB.

## Optimisation with guanidine chloride

The addition of guanidine chloride (GuHCl) in LAMP to increase assay kinetics has been previously demonstrated by *Zhang, Lowe & Gooding (2014)*. As such, we evaluated utilising GuHCl in the Str-LAMP assay to decrease the limit of detection (LOD), Tp, and inter-coefficient of variation (inter-CV). A concentration gradient ranging from 5 mM to 60 mM GuHCl (5 mM, 10 mM, 20 mM, 30 mM, 40 mM, 50 mM, and 60 mM) was performed on the Genie® II Machine (OptiGene Limited) as above, to determine the optimal concentration. Reactions contained 15 μl of ISO-DR004 (OptiGene Limited), 0.2 μM F3 and B3, 0.8 μM of FIP and BIP, 0.4 μM LB, and a specified concentration of GuHCl (above), with the synthetic positive control at $5.3 \times 10^5$ copies/μl ($1 \times 10^{-4}$ ng/μl) for each GuHCl concentration. The three concentrations which returned the fastest Tp where further assessed on the ability to amplify lower quantities of DNA, by testing concentrations of $5.3 \times 10^2$ copies/μl, 53 copies/μl, and 5.3 copies/μl ($1 \times 10^{-7}$ ng/μl to $1 \times 10^{-9}$ ng/μl) of the synthetic positive control. The GuHCl concentration which yielded the quickest and most consistent Tp was then determined to be the optimal final concentration.

## Str-LAMP limit of detection analysis and comparison

The Str-LAMP limit of detection (LOD) was performed with and without the addition of GuHCl to compare assay analytics. The synthetic positive control was serially diluted 10-fold from $5.3 \times 10^9$ copies/μl to 53 copies/μl (1 ng/μl to $1 \times 10^{-8}$ ng/μl) and evaluated in triplicate technical replicates across three assays with either 0 mM GuHCl or 20 mM GuHCl. All reactions were performed as above on the Genie® II machine (OptiGene Limited), with a fluorescence ratio threshold of 0.02 normalised fluorescence units. The LOD was determined as the lowest concentration of DNA which returned a Tp within 20 min, as per the manufacturer's guidelines (OptiGene). The inter-CV for each concentration

was calculated as the average Tp of each technical replicate across three assays (9 replicates per DNA concentration) multiplied by the average standard deviation (SD) for each replicate. To compare the assay's performance with and without GuHCl, the average Tp and standard deviation (expressed as minutes and seconds) difference, and the inter-CV difference was analysed using Microsoft Excel software (Microsoft Corporation, Redmond, WA, United States).

## Validation with clinical isolates

Clinical isolates of *S. equi* and *S. zooepidemicus* were kindly provided by ACE Laboratory Services (East Bendigo, Victoria, Australia), and Gribbles Veterinary Pathology (Clayton, Victoria, Australia). Isolates were received as culture swabs stored in AIMES transport media. The clinical isolates were grown on Columbian CNA agar supplemented with 5% sheep blood for 48 h at 37 °C, following which a single colony was resuspended in 200 µl of phosphate buffer saline (PBS: 137 mM NaCl, 2.7 mM $Na_2HPO_4$, 1.8 mM $KH_2PO_4$, pH 8.0) and boiled at 95 °C for 2 min. Each *S. equi* colony was amplified *via* PCR to determine successful growth. Reactions contained 12.5 µl of Go-Taq® Green master mix (Promega), 0.8 µM Str-LAMP F3 and B3 (Table 1, Set 2), 5 µl of sample or $dH_2O$ for a negative control, adjusted to 25 µl with $dH_2O$. Thermocycling conditions and results visualisation were performed as described above. Five microlitres of each crude sample were evaluated on the Str-LAMP assay in duplicates, as the same conditions above with a fluorescence threshold ratio of 0.01 normalised fluorescence units.

To further validate the specificity of the test, the Str-LAMP assay was evaluated against a range of other bacterial species. The bacteria included in this study were *E. coli* ($n = 6$), *Staphylococcus* spp. ($n = 3$), *Pseudomonas* spp. ($n = 3$), *Salmonella* spp. ($n = 3$), and *Yersinia* spp. ($n = 3$). Each bacterial species was resuspended in PBS and evaluated on the Str-LAMP assay as above.

The clinical sensitivity was calculated as the percentage of correct identification of *S. equi* within 20 min, and specificity was determined by the percentage of correct identification (*i.e.,* no amplification) of *S. zooepidemicus*. The correct classification ratio was evaluated as the percentage of total correct identifications of all bacterial species tested .

## RESULTS

### Primer design and optimisation

Of the three SeM primer sets evaluated, primer set 2 (Table 1) achieved the quickest and most consistent Tp. Interestingly, primer set 3 did not record a time for 1 ng/µl despite returning a positive result for 0.01 ng/µl (Fig. S1). During primer concentration optimisation of primer set 2, it was determined that combination 1 and 3 (Table 1) returned the fastest Tp (Table 2). As combination 3 returned a lower inter-CV (0.12%) it was decided to continue with this concentration. However, further analysis revealed the LOD to be relatively subpar at $1 \times 10^{-5}$ ng/µl (Data S1), therefore combination 1 was chosen as the optimal concentration instead.

**Table 2 Assessed primer concentrations and results for the Str-LAMP assay.**

| Primer combination | Primer concentration (F3/B3; FIP/BIP;LB µM) | Average Tp (mm:ss) | Average SD (mm:ss) | Inter-CV (%) |
|---|---|---|---|---|
| Combination 1 | 0.2; 0.8; 0.4 | 03:55 | 00:12 | 0.14 |
| Combination 2 | 0.2; 1.6; 2.0 | 06:17 | 00:07 | 0.14 |
| Combination 3 | 0.2; 0.8; 1.0 | 03:15 | 00:12 | 0.12 |

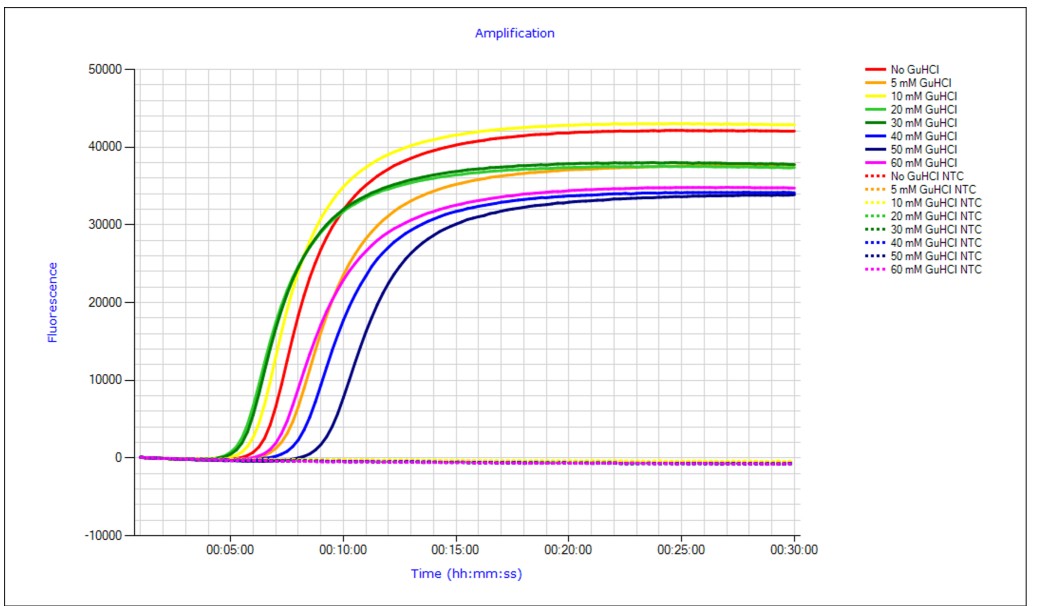

**Figure 2 Amplification curve plot for various final concentrations of guanidine hydrochloride (GuHCl) in the strangles loop-mediated isothermal amplification (Str-LAMP) assay.** Tested concentrations of GuHCl include 0 mM (red), 5 mM (orange), 10 mM (yellow), 20 mM (green), 30 mM (dark green), 40 mM (blue), 50 mM (dark blue), and 60 mM (purple). Each GuHCl had a no template control (NTC) represented in dotted lines of respective colours.

## Optimisation with guanidine hydrochloride (GuHCl)

Seven concentrations of GuHCl (5 mM, 10 mM, 20 mM, 30 mM, 40 mM, 50 mM, and 60 mM) were assessed using $5.3 \times 10^5$ copies/µl ($1 \times 10^{-4}$ ng/µl) of the synthetic positive control. Initial assessment revealed that final concentrations of 10, 20 and 30 mM returned the quickest Tp (Fig. 2). Following, we assessed 10, 20 and 30 mM of GuHCl ability to detect lower concentrations of DNA. Whilst each concentration was able to amplify 5.3 copies/µl ($1 \times 10^{-9}$ ng/µl) within 20 min, it was determined a final concentration of 20 mM GuHCl had the most consistent and fastest Tp (Table 3). Thus, this concentration of GuHCl was used in subsequent assays.

## Str-LAMP LOD analysis and comparison

To determine the effectiveness of GuHCl we compared the Str-LAMP LOD using no GuHCl and 20 mM GuHCl. Without GuHCl the Str-LAMP's LOD was $5.3 \times 10^2$ copies/µl ($1 \times 10^{-7}$ ng/µl) within the 20-minute threshold, with an average inter-CV at 4.55%.

**Table 3** Limit of detection (LOD) time to positive (Tp) results for 0 mM, 10 mM, 20 mM, and 30 mM GuHCl using the synthetic positive control.

| | Tp (mm:ss) | | | |
|---|---|---|---|---|
| Sample concentration (copies/µl) | 0 mM GuHCl | 10 mM GuHCl | 20 mM GuHCl | 30 mM GuHCl |
| $5.3 \times 10^2$ $(1 \times 10^{-7}$ ng/µl) | 12:47 | 10:06 | 09:46 | 10:37 |
| 53 $(1 \times 10^{-8}$ ng/µl) | 15:32 | 12:40 | 11:30 | 14:18 |
| 5.3 $(1 \times 10^{-9}$ ng/µl) | 25:58 | 14:08 | 12:43 | 12:05 |

**Table 4** Comparison of limit of detection (LOD), time to positive (Tp) and inter-coefficient of variation (CV) between 0 mM GuHCl and 20 mM GuHCl in the Str-LAMP assay.

| | 0 mM GuHCl | | | 20 mM GuHCl | | |
|---|---|---|---|---|---|---|
| Concentration (ng/µl) (copies/µ) | Av Tp[a] (mm:ss) | Av SD[b] (mm:ss) | Inter-CV (%) | Av Tp[a] (mm:ss) | Av SD[b] (mm:ss) | Inter-CV (%) |
| $1 \times 10^{-0}$ $(5.3 \times 10^9)$ | 05:36 | 00:58 | 0.94 | 0:2:50 | 00:57 | 0.47 |
| $1 \times 10^{-1}$ $(5.3 \times 10^8)$ | 06:52 | 01:06 | 1.31 | 03:48 | 00:19 | 0.21 |
| $1 \times 10^{-2}$ $(5.3 \times 10^7)$ | 08:15 | 00:58 | 1.38 | 04:44 | 00:46 | 0.49 |
| $1 \times 10^{-3}$ $(5.3 \times 10^6)$ | 10:16 | 02:40 | 4.75 | 06:17 | 00:38 | 0.69 |
| $1 \times 10^{-4}$ $(5.3 \times 10^5)$ | 11:11 | 00:14 | 0.45 | 07:36 | 00:44 | 0.97 |
| $1 \times 10^{-5}$ $(5.3 \times 10^4)$ | 11:49 | 00:56 | 1.91 | 08:17 | 00:07 | 0.17 |
| $1 \times 10^{-6}$ $(5.3 \times 10^3)$ | 12:44 | 00:43 | 1.58 | 09:35 | 00:12 | 0.33 |
| $1 \times 10^{-7}$ $(5.3 \times 10^2)$ | 19:02 | 04:52 | 16.08 | 10:58 | 00:31 | 0.98 |
| $1 \times 10^{-8}$ $(5.3 \times 10^1)$ | 21:52 | 03:19 | 12.59 | 13:19 | 01:42 | 3.93 |
| $1 \times 10^{-9}$ $(5.3 \times 10)$ | NA[c] | NA | NA | INC[d] | INC | INC |

**Notes.**
[a] Average time to positive (Tp) across three assays with triplicate replicates
[b] Average standard deviation (SD) across three assays with triplicate replicates.
[c] NA, no amplification
[d] INC, inconsistent results

However, it should be noted the individual inter-CV for $5.3 \times 10^2$ copies/µl was calculated at over 16% (Table 4), with large variabilities in Tp across replicates. The Str-LAMP assay with the addition of 20 mM GuHCl allowed for a 10-fold decrease of LOD, at 53 copies/µl $(1 \times 10^{-8}$ ng/µl) within an average of about 13 min and 20 s, and an inter-CV ranging between 0.17% to 3.93%. Comparatively, for 53 copies/µl the inter-CV was calculated at 3.39%, compared to 12.9% without GuHCl (Table 4). Although the Str-LAMP assay with GuHCl was occasionally able to amplify 5.3 copies/µl $(1 \times 10^{-9}$ ng/µl), this result w2as not consistently achieved across all assays (Table 4). Therefore, it is not being reported as the final LOD. On average, GuHCl decreased the Str-LAMP assay Tp by approximately 4 min and 30 s. At lower concentration's $(5.3 \times 10^2$ copies/µl and $5.3 \times 10^1$ copies/µl) Tp decreased by over 8 min (Table 4). The average decrease in SD between no GuHCl and 20 mM GuHCl was calculated to be 1 min and 7 s (Fig. 3). Overall, GuHCl decreased the inter-CV by an average of 4%, with $5.3 \times 10^2$ copies/µl individual inter-CV decreasing by 15.10% (Table 4)

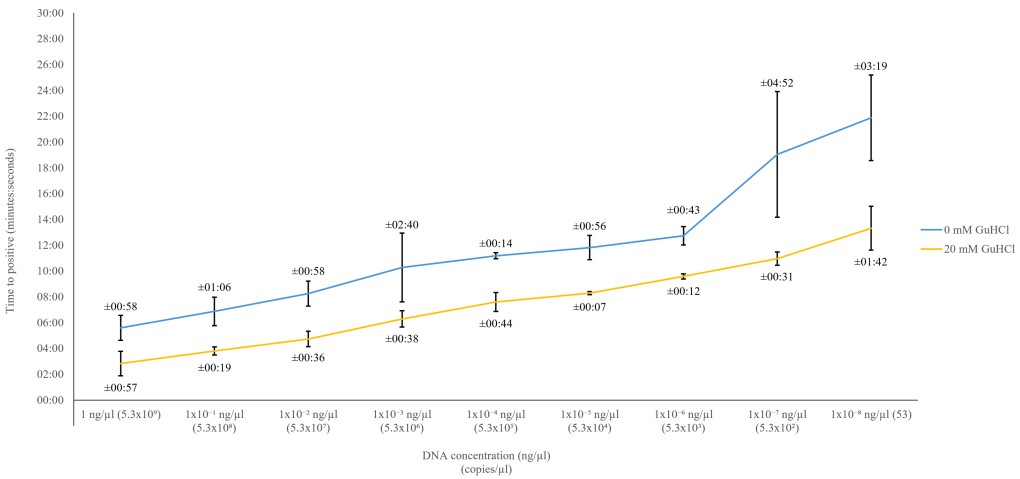

**Figure 3  Comparison of 0 mM GuHCl and 20 mM GuHCl.** Comparison of the limit of detection (LOD) and standard deviation (SD) between technical replicates, of 0 mM GuHCl (grey stripes) and 20 mM GuHCl (solid grey). Error bars represent the variation between replicate concentration's Tp (mm:ss), average SD across replicates is stated above.

**Table 5  Str-LAMP identification results for *Streptococcus equi* subspecies *equi* and *S. equi* subspecies *zooepidemicus* clinical isolates, and other bacterial species.**

| Bacteria | Str-LAMP positive (#) (%) | Str-LAMP negative (#) (%) | Total |
|---|---|---|---|
| *S. equi* subsp. *equi* | 94 [91.3] | 9 [8.7] | 103 |
| *S. equi* subsp. *zooepidemicus* | 3 [11.1] | 24 [88.9] | 27 |
| *E. coli* | 0 [0] | 6 [100] | 6 |
| *Staphylococcus* spp. | 0 [0] | 3 [100] | 3 |
| *Pseudomonas* spp. | 0 [0] | 3 [100] | 3 |
| *Salmonella* spp. | 0 [0] | 3 [100] | 3 |
| *Yersinia* spp. | 0 [0] | 3 [100] | 3 |
| Total | 98 | 33 | 148 |

## Validation of the Str-LAMP with clinical isolates

In total 148 samples, comprised of *S. equi* ($n = 103$), *S. zooepidemicus* ($n = 27$), *E. coli* ($n = 6$), *Staphylococcus* spp. ($n = 3$), *Pseudomonas* spp. ($n = 3$), *Salmonella* spp. ($n = 3$), and *Yersinia* spp. ($n = 3$), were evaluated on the Str-LAMP assay. Of the 148 samples, 136 samples were correctly identified as either positive or negative, resulting in a correct classification ratio of 91.8% for the Str-LAMP assay. A total of 94 of *S. equi* samples returned positive results, yielding clinical sensitivity calculated at 91.3% (Table 5). However, it should be noted that 6 of the 9 (66%) *S. equi* samples that returned false negative results by the Str-LAMP assay was due to amplification occurring beyond the 20-minute threshold (Data S2). The average Tp for *S. equi* samples was approximately 13 min and 20 s (Fig. 4). Of the 27 *S. zooepidemicus* samples, the Str-LAMP assay correctly identified 24 samples as negative. All other tested bacterial species were deemed negative on the Str-LAMP assay (Fig. S2), resulting in a specificity of 93.3% (Table 5).
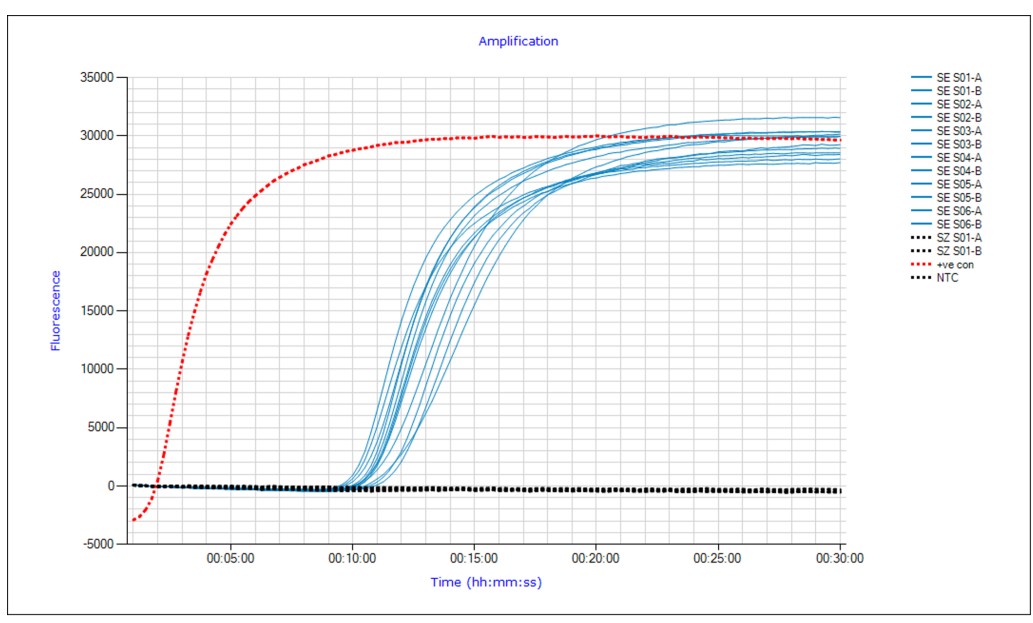

**Figure 4** **Amplification curve plot of the strangles loop-mediated isothermal amplification (Str-LAMP) assay using clinical isolates.** Clinical isolates of *Streptococcus equi* subspecies *equi* (SE –blue) and *S. subspecies zooepidemicus* (SZ –Black dotted lines) were tested on the Str-LAMP assay, with SE all returning positive results within 20 min, and no amplification of SZ. The synthetic positive control (+ve con) is represented in red, and the no template control (NTC) is denoted in black.

## DISCUSSION

This study presents a rapid and highly repeatable LAMP assay for the detection of the SeM gene in *S. equi*. While previous studies have noted variation of the SeM gene across multiple *S. equi* strains, we targeted a conserved region, which shows limited homology to the closely related pathogen *S. zooepidemicus* (*Mitchell et al., 2021*; *Rotinsulu et al., 2023*) (Fig. S3). This specificity was demonstrated through validation with clinical isolates, showing a high discrimination ability.

Strangles is considered the most contagious equine respiratory disease due to the transmissible state of *S. equi*, resulting in immense difficulty to overcome outbreaks in equine facilities (*Boyle, 2023*; *Rendle et al., 2021*). Therefore, it is crucial to develop an accurate and rapid surveillance technique for the detection of strangles in horses (*Taylor & Wilson, 2006*). While multiple PCR-based methodologies have been developed, a nationally standardised diagnostic test for *S. equi* remains eluded resulting in variation amongst results (*Boyle, 2023*). Yet, a qPCR assay developed by *Båverud, Johansson & Aspan (2007)* is currently commercially available in North America (*Boyle et al., 2018*; *McGlennon et al., 2021*). Whilst the specificity of this assay is reportedly high at 97% (*Båverud, Johansson & Aspan, 2007*; *Boyle, Stefanovski & Rankin, 2017*), multiple efforts have been attempted to improve the clinical sensitivity, which remains unreported, to no avail (*Boyle et al., 2018*). As such, it is apparent a clinically and analytically sensitive surveillance methodology is vital.

The Str-LAMP assay presented in this paper begins to address this need. The assay produces rapid results while maintaining excellent analytical sensitivity, with the ability to detect 53 copies/μl of synthetic DNA within an average of 13 min and 20 s. Furthermore, this assay maintains a good sensitivity and specificity of 91.3% and 93.3%, respectively, when testing clinical isolates of *S. equi* in parallel with the closely related *S. zooepidemicus*. As strangles is a notable disease of concern in horses, there have been multiple previous attempts of developing a reliable LAMP assay (*Boyle, Stefanovski & Rankin, 2017*; *Boyle et al., 2021*; *Hobo, Niwa & Oku, 2011*). *Hobo, Niwa & Oku (2011)* demonstrated the first strangles LAMP assay also targeting the SeM gene, showing a high clinical sensitivity and specificity at 100% each. However, the assay requires 43 min to achieve these results (*Hobo, Niwa & Oku, 2011*). Whilst this is an improvement compared to a qPCR assay (*Båverud, Johansson & Aspan, 2007*), which requires over an hour for amplification, our assay can be completed in less than one-third of the time. Likewise, *Boyle, Stefanovski & Rankin (2017)* demonstrated a LAMP assay for strangles, purportedly achieving 100% clinical sensitivity and specificity within 30 min. However, it is worth noting that upon re-evaluation, the clinical sensitivity and specificity reportedly decreased to 92% and 71%, respectively (*Boyle et al., 2021*). This could suggest the assay developed by *Boyle, Stefanovski & Rankin (2017)* may lack repeatability, rendering variation in results likely. Contrarily, the presented Str-LAMP assay results were highly consistent throughout testing, with the inter-CV ranging from 0.17% to 3.93%, giving an average of just 0.92% across multiple assays.

The consistency of the Str-LAMP assay presented here can likely be attributed to the use of GuHCl. It has been previously demonstrated that GuHCl can decrease reaction speed whilst concurrently decreasing the LOD and clinical sensitivity (*Zhang et al., 2020*). In fact, our study found a final concentration of 20 mM GuHCl decreased the Str-LAMP assay LOD by 10-fold, compared to without GuHCl. Utilising this chemical additive allowed for the clinical sensitivity of the assay tested with concentrations of 530 copies/μl and 53 copies/μl to increase to 100% each from 78% and 22%, respectively. These results are similar to those found by *Zhang et al. (2020)* where the sensitivity of their assay measuring 100 copies of SARS-CoV-2 increased from under 50% to over 90%, and 50 copies to increase from 30% to 70%. However, multiple repeated studies unanimously found a final concentration of 40 mM was optimal for their respective assays (*Anastasiou et al., 2021*; *Hewadikaram et al., 2022*; *Zhang et al., 2020*), which is twice the amount we observed. As these assays were performed using reverse-transcription LAMP (RT-LAMP) which requires an additional enzyme compared to LAMP (*Knox & Beddoe, 2021*), it is plausible to hypothesise that GuHCl may interact with the polymerase. However, *Zhang et al. (2020)* noted there was no variation of results when utilising different polymerases during developmental stages, thus inferring there is no interaction with GuHCl. Yet, it should be noted that majority of these assays are performed on SARS-CoV-2 RNA, thus it is possible GuHCl is somewhat sequence-dependent. GuHCl is often utilised in amplification techniques, such as PCR, due to its chaotropic properties to denature nucleic acid and hinder secondary and tertiary structure formation (*Lewin & Munroe, 1965*; *Tanford, 1970*). Through its disruption of hydrogen bonding, GuHCl disturbs the structured arrangement of water

molecules surrounding nucleic acid, thereby increasing the solubility of hydrophobic surfaces (*Tanford, 1970*). This facilitates denaturation of double-stranded DNA and assists its transition to a single-stranded state (*Taylor et al., 1995*). This in turn allows for the formation of stable primer-template duplexes, enabling a more efficient primer binding (*Zhang et al., 2020*). As AT-base pairs are exposed to more hydrophobic surfaces (*Melchior & Von Hippel, 1973*), it is conceivable that a difference in AT and GC content, thus a difference in target sequence, could require more or less concentration GuHCl. As such, GuHCl potentially works alongside with LAMP polymerases in strand displacement, rather than direct interaction (*Zhang et al., 2020*). Yet, the actual mode of mechanism of GuHCl remains unknown in LAMP due to limited studies, necessitating the need for further investigation.

The Str-LAMP assay was validated using clinical isolates of *S. equi* and *S. zooepidemicus*, as well as other bacterial species. Out of 148 tested specimens, our assay was able to correctly identify 136 samples (91.8%). This is in contrast to the assay developed by *Boyle, Stefanovski & Rankin (2017)* which had reported a correct classification of 75%. Interestingly, the Str-LAMP average Tp for *S. equi* clinical isolates was 13 min and 20 s, identical to that found when utilising the synthetic positive control during optimisation stages. Of the 9 *S. equi* samples which were deemed negative, three samples (~33%) returned no amplification while six (~67%) samples still amplified but were beyond the 20 min threshold. This coupled with the parallel average Tp could suggest these six samples may be outside the Str-LAMP assay's LOD range, demonstrating some limitations in our assay. However, as *Hobo, Niwa & Oku (2011)* reported, minute amounts of DNA quantities would unlikely pose a concern in a clinical setting, as strangles-infected equine are likely to be shedding bacterial DNA at a substantially higher load. Yet, the Str-LAMP LOD would require re-evaluation after optimisation into sample extraction procedures. Whilst we utilised a simple sample extraction method, this was performed on culture colonies of clinical isolates, which may be considered as relatively pure. Previous studies have demonstrated the sample site and procedure for strangles can greatly affect the performance of a diagnostic procedure (*Boyle, Stefanovski & Rankin, 2017*; *Lindahl et al., 2013*). *Lindahl et al. (2013)* found using nasopharyngeal lavage achieved the greatest clinical sensitivity for both culture and PCR, with the use of a dry cotton swab preferential. In contrast, *Boyle, Stefanovski & Rankin (2017)* reported that guttural pouch lavage samples had superior sensitivity compared to nasopharyngeal-derived samples when optimising their strangles assay. Although both methods achieved satisfactory results, it's worth noting a lavage can be somewhat invasive compared to alternative methods. *Kinoshita et al. (2016)* demonstrated a simple 10 min boiling step of tracheal washes was able to consistently detect *Rhodococcus equi* in their LAMP assay. However, the efficiency of boiling samples has been debated, citing heat-treatment does not efficiently denature DNases leading to false-negative results (*Kinoshita, Niwa & Katayama, 2014*). Yet, the use of GuHCl in the Str-LAMP assay would be able to compensate for this. Furthermore, the convenience of heat-treatment as an extraction protocol should not be overlooked, particularly for LAMP which is typically tolerant of inhibitors (*Kaneko et al., 2007*).

Throughout our study we utilised the Genie® II machine (OptiGene Limited) to perform the Str-LAMP assay. While this machine is an invaluable asset to LAMP that can be performed in-field, it would not be feasible to disperse machines across a large number of equine farms. As LAMP has garnered a substantial amount of research interest, particularly for the detection of equine diseases, field-deployable result outputs are continuing to evolve. *Boyle et al. (2021)* demonstrated such advancements when incorporating their strangles LAMP assay into a previously developed microfluidic device (MFD), deemed Smart Cup (*Liao et al., 2016*). The Smart Cup, which utilises a Thermos bottle as the chamber, was capable of providing real-time fluorescence monitoring through a smartphone's camera. However, despite the perceived simplicity, an intricate component of this device was the use of a microfluidic "lab-on-a-chip" (LOC) (*Boyle et al., 2021*; *Liao et al., 2016*). Although utilising a LOC might be straightforward for a trained laboratory worker, it could present complexity for individuals without specific training, thereby limiting its accessibility and usability. We suggest integrating our Str-LAMP assay into a lateral flow device (LFD) output, a now universally recognised diagnostic technique (*Mak et al., 2020*; *Xie et al., 2022*). LFDs have become increasingly prevalent for the self-detection of SARS-CoV-2, through rapid antigen testing. Furthermore, these LFDs have also been frequently coupled with LAMP assays to produce unanimous results (*Knox, Zerna & Beddoe, 2023*). While this may necessitate additional optimisation and validation of the Str-LAMP assay, previous studies have observed minimal discrepancies in assay analytics when adapting to an LFD result output system (*Zhang, Lowe & Gooding, 2014*). Therefore, the advantage of on-farm self-surveillance for strangles using the Str-LAMP assay with an LFD could significantly contribute to rapid outbreak containment.

## CONCLUSIONS

We have developed a highly rapid and analytically sensitive assay for the detection of strangles, with immense reproducibility. The positioning of the primers to target the SeM gene outside regions susceptible to sequence variation, and little homology to *S. zooepidemicus*, enables for specific targeted detection of strangles. Our assay was able to detect 53 copies/µl of *S. equi* synthetic DNA within 13 min and 20 s. Furthermore, the Str-LAMP assay demonstrates this efficiency through the use of clinical isolates, obtaining a high clinical sensitivity of 91.3% and specificity of 93.3%. The observed improvements and stabilisation attribute with the use of GuHCl offers further insights into the beneficial application of chemical additives, suggesting avenues for future research. Although further validation is necessary for sampling, the assay's repeatability and accuracy provides a means for non-trained personnel to conduct the assay in-field. With its potential to significantly enhance strangles surveillance across farms, our Str-LAMP assay can aid in the impediment of widescale outbreaks and associated financial burdens.

## ACKNOWLEDGEMENTS

We would like to thank ACE Laboratory Services, and Gribbles Veterinary Pathology for access to clinical samples.

### Funding

This research was supported by the Cooperative Research Centres Project (CRC-P) awarded to GeneWorks and La Trobe University. Alexandra Knox is supported by a La Trobe Industry Ph.D. scholarship and the Defence Science Institute, an initiative of the State Government of Victoria. The funders had no role in study design, data collection and analysis, decision to publish, or preparation of the manuscript.

### Grant Disclosures

The following grant information was disclosed by the authors:
The Cooperative Research Centres Project (CRC-P) awarded to GeneWorks and La Trobe University.
A La Trobe Industry Ph.D. scholarship.
Defence Science Institute, an initiative of the State Government of Victoria.

### Competing Interests

Travis Beddoe is an Academic Editor for PeerJ.

### Author Contributions

- Alexandra Knox conceived and designed the experiments, performed the experiments, analyzed the data, prepared figures and/or tables, authored or reviewed drafts of the article, and approved the final draft.
- Travis Beddoe conceived and designed the experiments, authored or reviewed drafts of the article, and approved the final draft.

### Data Availability

The raw measurements of PCR and LAMP run are available in the Supplementary File.

### Supplemental Information

Supplemental information for this article can be found online at http://dx.doi.org/10.7717/peerj.17955#supplemental-information.

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
