# Peer review of "Enhancement of loop-mediated isothermal amplification (LAMP) with guanidine hydrochloride for the detection of Streptococcus equi subspecies equi (Strangles)"

_PeerJ, doi:10.7717/peerj.17955_

## Round 0.1 · original submission · Major Revisions

Please provide a point-by-point response to all the reviewers' comments.

Reviewer 1 ·

Basic reporting

This study is preliminary to test their hypothesis about developing and improvement of LAMP to detect S. equi subsp. equi (SEE). It is a preliminary descriptive study that does not apply to the real world (on-field deployment).
I have some questions/comments for the authors to improve their manuscript.
1. The introduction and Discussion are too long, I recommend the authors shorten this part. Please mentioned to your result with other studies.
2. Cross-reactivity should be conducted with additional bacterial species, although the authors have tested with closely related SEE (SEZ) but I feel it is not enough.
3. Line 276-280, I am not clear about amount of isolates used in this study. The total is 130 isolates, but 118 were positive for SEE or SEZ (you mentioned 103 SEE and 27 SEZ). However, your study wants to detect only SEE. That means some SEZ give positive results, is it true? If so, this LAMP is not 100% specificity (cross-reactivity). Please clear this part and also please discuss it.
4. Line 279-280: You have mentioned 94 positives, 6 were also positive (maybe later). What about the rest 3 isolates? Are they completely negative?
5. I suggest the author to calculate percentages of sensitivity, specificity, positive-predictive value, and negative-predictive value. Although you did not compare LAMP with gold standard/reference method in the current study, however, the bacterial isolates used in this study have been identified based on the gold standard/reference method (culture). I think that it is possible to calculate it.

Experimental design

Experimental design is appropriate to research questions and objective of this study. However, some additional experiments are required.

Validity of the findings

Valid and according to the study design.

·

Basic reporting

No comments

Experimental design

No comments

Validity of the findings

No comment

Additional comments

1. Provide a standard curve for fluorescence vs DNA concentration to identify trends in Tp values. Standard curves represent any drastic difference in Tp values within technical replicates, especially among those with low concentration.
2. Why were 6 S.equi samples had high Tp times? Were the concentrations of purified DNA low? If so, mentioning them will enhance the study interpretations.
3. What was the Tp for 3 S. zooepidemicus samples that were positive by LAMP assay? Were they slightly lower than 20?
4. How do the authors justify amplification of 3 S. zooepidemicus samples and non-amplification in 3 equi samples? How much was the DNA concentration for these samples? Were there any homologous regions or were the locus of interest sequenced? If so, share the relevant data.
5. The least dilution was 53 copies/ul for calculating LOD, and authors report this is the analytical LOD. How did the authors arrive at such conclusion without testing concentrations lower than this? Or were they tested, and Tp > 20?
6. As OptiGene mastermix has fluoroscent dye (FAM-channel), were melt curves estimated? Share the relevant images if available.
7. Figure 3 can be better represented with line plot (with mean and standard errors) as there appears to be a sudden increase in reactions without GuHCl after 10-6 ng/ul.
8. For the LOD identified, with or without GuHCl, the inter-CV is ~4-12%, but the abstract mentions 0.92%. It would be better if the range of inter-CV for highest and lowest concentrations is mentioned in the abstract itself.
9. At what DNA and GuHCl concentrations the inter-CV is 0.92% as mentioned in the abstract?

Reviewer 3 ·

Basic reporting

The manuscript describes the development and validation of a LAMP assay for rapid detection of S. equi, bacterium responsible for strangles in horses. Furthermore, the authors describe how the addition of guanidine hydrochloride to the LAMP reaction can enhance the assay’s performance and detection capabilities.

Introduction
Line 74: An infected horse can quickly become a chronic asymptomatic carrier which “may” periodically spread the bacterium, “and become” a source of new or “contribute” recurrent outbreaks.
Line 76: Predisposing factors of what? It is not clear what the author is referring too. Please clarify if you are referring to predisposing factors to infection.
Line 78: It is recommended to list the Australian states strangles is notifiable, rather than state it is notifiable in some states.
Line 85: estimated some outbreaks “involving” hundreds of horses,
Line 87: I suggested including the bacterium responsible for disease; “Whilst S. equi induced disease can occur at any age, the most severe “clinical presentations” are often seen in younger horses…..
Line 89: “Infected horses” typically present with sudden pyrexia as the first sign,
Line 97: Does viable mean infectious? When discussing bacteria, I would suggest using the term infectious instead of viable.
Line 98: Disease is not transmissible, but the bacterium causing disease is. Double check your terminology.
Line 101: disease surveillance to inhibit the “transmission of the bacterium”……
Line 102: Historically, “microbial” culture has been the gold-standard detection method for strangles.
Line 104: Subsequently, diagnostic approaches for “detection of” strangles are favoured towards nucleic acid amplification techniques….. are these diagnostic approaches detecting strangles or S. equi? Again, terminology.
Line 108: However, there is no “nationally standardised” procedure regarding this diagnostic “test”, therefore, results may vary across laboratories.
Line 113: It is recommended to provide additional information on LAMP, such as the use of a polymerase with high strand displacement activity in addition to a replication activity etc.
Line 127: Considering you are applying this technology to horses, shouldn’t it be “stable-side’ instead of “pen-side”?

Materials and Methods
Line 140: Three sets of primers were designed to “target” different regions of the S. equi-specific M protein……Are you targeting a protein or a gene?
Line 165: Can the author describe the acronyms of the primers F3/B3, FIP/BIP and LB? Why isn’t there a LF? I’m assuming LB is a loop primer?
Line 167: Reactions were performed in “triplicate”……
Line 170: I would suggest rewording “Reactions were considered positive if product amplification was achieved within 20 min (which raises the question of why run amplification for 60 min? What happens if there is amplification after 20 min? Are these samples considered negative? Please clarify). You might also want to mention anneal temperature. Each LAMP target will have its own signature anneal temperature which is determined by the GC content of the generated amplicon. Positive amplification followed by an anneal temperature at the expected temperature is confirmation of a true positive in a sample.
Line 186: was performed on the Genie® II Machine (OptiGene Limited) as above, to determine the optimal concentration.
Line 199: and evaluated in triplicate technical replicates across three assays …..

Results
Line 269: GuHCl decreased the Str-LAMP assay Tp by roughly approximately 4 minutes and 30 seconds.

Discussion
Line 305: See my comment above regarding nationally standardised diagnostic test for S. equi.
Line 317: testing clinical isolates of S. equi alongside in parallel with the closely related S. zooepidemicus
Line 349: I would like to have read a brief description, similar to what the author has described in the Discussion, of why GuHCl was selected in the Introduction.

Experimental design

No comment

Validity of the findings

No comment

---

## Round 0.2 · accepted · Accept

The authors have addressed all the comments. Two reviewers who accepted to review the revised version of the manuscript have endorsed the publication of the manuscript. The manuscript is now ready for publication.

Reviewer 1 ·

Basic reporting

The author has responded my comments and questions clearly. I have no additional concerns.

Experimental design

The author has responded my comments and questions clearly. I have no additional concerns.

Validity of the findings

The author has responded my comments and questions clearly. I have no additional concerns.

Additional comments

The author has responded my comments and questions clearly. I have no additional concerns.

Reviewer 3 ·

Basic reporting

No comment

Experimental design

No comment

Validity of the findings

No comment

Additional comments

I am satisfied with the revisions the author has made to the manuscript.